# An Efficient Contrastive Unimodal Pretraining Method for EHR Time Series Data

Ryan King, Shivesh Kodali, Conrad Krueger, Tianbao Yang, and Bobak J. Mortazavi

*Computer Science & Engineering*, *Texas A&M University*, College Station, United States

{kingrc15, shivesh_2001, conradk1234, tianbao-yang, bobakm}@tamu.edu

*Abstract*—**Machine learning has revolutionized the modeling of clinical timeseries data. Using machine learning, a Deep Neural Network (DNN) can be automatically trained to learn a complex mapping of its input features for a desired task. This is particularly valuable in Electronic Health Record (EHR) databases, where patients often spend extended periods in intensive care units (ICUs). Machine learning serves as an efficient method for extract meaningful information.**

**However, many state-of-the-art (SOTA) methods for training DNNs demand substantial volumes of labeled data, posing significant challenges for clinics in terms of cost and time. Self-supervised learning offers an alternative by allowing practitioners to extract valuable insights from data without the need for costly labels. Yet, current SOTA methods often necessitate large data batches to achieve optimal performance, increasing computational demands. This presents a challenge when working with long clinical timeseries data.**

**To address this, we propose an efficient method of contrastive pretraining tailored for long clinical timeseries data. Our approach utilizes an estimator for negative pair comparison, enabling effective feature extraction. We assess the efficacy of our pretraining using standard self-supervised tasks such as linear evaluation and semi-supervised learning. Additionally, our model demonstrates the ability to impute missing measurements, providing clinicians with deeper insights into patient conditions.**

**We demonstrate that our pretraining is capable of achieving better performance as both the size of the model and the size of the measurement vocabulary scale. Finally, we externally validate our model, trained on the MIMIC-III dataset, using the eICU dataset. We demonstrate that our model is capable of learning robust clinical information that is transferable to other clinics.**

*Index Terms*—**EHR Time Series, Unimodal Pretraining, Contrastive Pretraining, Masked Pretraining**

**E**LECTRONIC health records (EHRs) contain a wealth of information about patient outcomes and treatment effects. Leveraging large amounts of EHR data holds immense potential for various healthcare applications, including clinical decision support, predictive modeling, personalized medicine, epidemiological studies, and healthcare resource optimization. These datasets offer insights into disease progression, treatment effectiveness, adverse events, and population health trends. However, finding useful patterns in these databases poses a significant obstacle [1] due to the large amounts of data and limited resources needed to extract insights.

Deep Neural Networks (DNNs) have proven to be a powerful tool for learning complex patterns from large amounts of labeled data. Advances in novel model architectures such as transformers [2], ViTs [3], and ResNet [4] have enabled models to learn more complex and robust task features. However, these methods require substantial amounts of labeled data, a resource-intensive and costly endeavor for clinicians and clinics.

Recently, self-supervised learning has relieved some of the requirements for large amounts of labeled data by devising methods for learning from only the input data. Each self-supervised method develops a proxy task to train a model. These methods usually involve the application of a random transformation to an image and then a prediction related to the transformation such as rotation prediction [5], colorization [6], or in-painting [7]. Amongst the methods, masked token prediction, from natural language processing (NLP), and contrastive learning have emerged as high performing pretraining methods with some pretrained models achieving nearly the same performance as completely supervised methods. Masked token prediction models, apply a random masking to the input data and train the model to predict the masked value. The objective of contrastive pretraining attempts to maximize the similarity between the learned embeddings of positive pairs. Contrastive pretraining of medical time series data was recent proposed in [8]. However, many of these methods have large computational requirements which can be be exacerbated by the already large computational requirements necessary for performing operations on long time series data.

In this paper, we develop a pretraining method that is capable of handling large time series data while accounting for large batches necessary for contrastive pretraining. Our method utilizes a modified triplet embedding, first proposed in [9]. Similar to [10], this embedding allows us to treat our triplets like tokens in Natural Language Processing (NLP). We are then able to break our method up into two parts: a triplet level task and a sequence level task. For our triplet level task, we perform a modified version of masked pretraining by masking the value of the measurements. Our model is then trained to predict the missing value. For our sequence level task, we utilize a contrastive objective. Unlike contrastive methods found in [11], [12], we use a smaller batch size. To account for the negative pairs needed to make contrastive learning more efficient we utilize a gradient estimator of the contrastive term.

To evaluate the effectiveness of our pretraining, we train a model with our proposed objective. We then evaluate that model on the common linear evaluation and semi supervised setting using 2 downstream tasks: in-hospital mortality and phenotyping. We further show the utility of our model when imputing missing measurements. We provide our model with a

set of query times and measurements and compare the models predicted measurement value with a ground truth value. We see that our model is capable of imputing reasonable measurement values as indicated by a Normalize Mean Squared Error of 0.409. We further evaluate the ability of our model to scale up in the number of measurement features. We see an increase in model performance as the number of measurement features increases as measured by our linear evaluation. Finally, we show that the features learned by our model during pretraining are robust across different clinics. We pretraining our model using the MIMIC-III dataset and externally validate it on the eICU dataset using a linear evaluation. In doing so, we simulate the transfer of a model, trained on data from a large clinic, to a group of smaller more diverse clinics. We summarize our contributions as follows:

1) We develop a pretraining method that combines sequence-level pretraining tasks (contrastive learning) and token-level tasks (masked imputation). This approach efficiently handles long sequences of time series.
2) We show that our pretraining is an effective method for learning general information about EHR dataset by evaluating the ability of our pretrained model to impute data and predict patient outcomes with both in-distribution and out-of-distribution data.
3) We demonstrate the ability of our model and pretraining to learn better representations when the number of measurements features is increased during pretraining.

## I. RELATED WORKS

### A. EHR Modeling

Modeling EHR data poses several challenges due to irregular sampling of measurements and missing values. [13] addresses the issue of irregular sampling by averaging data per hour. This transforms the irregular time series into uniform intervals of measurement. However, by averaging hourly data, they loss information about frequent measurements that can occur when patients are in critical states. In addition, less frequent measurement may not be taken hourly leaving missing values in their data. To handle the missing value issue here, they propose imputing using recent values where available or "normal" value otherwise. This provides their model with reasonable information to make outcome predictions. However, there are numerous issues with this imputation strategy. Firstly, patient information does not remain the same at each time step until this next measurement is taken. Second, imputing "normal" values where no measurements are taken could bias the model to an undesirable outcome. Lastly, by formulating the measurement time series in this way, we are required to include a dense representation of the measurement time series for every measurement we care about. We believe that this method include redundant information and limits that ability of practitioners to include more diverse measurements.

Instead, [9] proposes using a triplet embedding to address the issue of irregularly sampled data and missing values. In this method, each measurement is represented as a triplet of the measurement type, value, and time. These three features then undergo separate embeddings before being added together to receive a final representation of the triplet. A model is then pretrained to forecast future values before being used for downstream tasks.

There are many advantages to the triplet embedding in [9] over the dense imputation strategy listed above. One being that we do not need to deal with missing data or irregular sampling. Additionally, since we take measurements as they come, we can include any number of measurements without the enormous computational requirement necessary with dense imputation. That is why we decide to use this type of embedding in our work.

### B. Self-Supervised Learning

Self-Supervised Learning is the task of learning from unlabeled data using proxy tasks. There are various proxy tasks that can be used such as rotation prediction [5], in-painting [7], and colorization [6]. Each of these tasks requires a model to learn from available data to correctly complete the task.

Masked token prediction [10] has emerged as the state-of-the-art method for pretraining text models in NLP. This method tasks, as it's input, a series of token representing a sentence. Tokens are masked at random using a special masked token and fed to a target model with the goal of predicting the token that was masked. In addition, a class token is append to the sequence and used to predict a separate proxy task such as next sentence prediction.

While this method may seem unrelated to EHR time series, we show later that by utilizing a triplet embedding, we are able to treat each measurement in our time series as a token. This allows us to leverage masked token prediction for pretraining.

### C. Joint Embedding Self-Supervised Learning

Joint Embedding Self-Supervised Learning (JE-SSL) is an type of SSL which attempts to learn from unlabeled data by maximizing the similarity between two positive instances. JE-SSL starts by constructing a set of positive pairs. For example, these pairs can be two random augmentations of a single input [12], [14] or two modalities with information about the same event [8], [11]. The notion of positive pairs can change depending on the domain but the intuition is that positive pairs represent the same instance. So, given a positive pair as the input to an encoder, the goal of JE-SSL is to maximize the similarity between the outputs or embeddings of the pairs.

However, simply maximizing the similarity between two positive pairs can lead to a phenomenon called dimensional collapse [15] where encoders can maximize the similarity between all positive pairs by outputting a trivial vector such as the zeros vector. Different methods have been proposed for avoiding this collapse. Non-contrastive methods, such as [14] propose minimizing the euclidean distance between embeddings while whitening the embedding space. Contrastive pretraining methods [8], [11], [12] propose using the InfoNCE loss with maximizes the cosine similarity between the embeddings of positive pairs, while minimizing the similarity between negative pairs.

In [8], EHR time series and clinical notes are used to train two modality specific encoders using a modified bimodal contrastive loss. While they show that their pretraining is capable of learning meaningful representations for downstream tasks, they require paired measurements and notes which may not be available in all EHR datasets. Additionally, they use a smaller batch size for their pretraining due to the need to train multiple encoders. Our method only requires measurement data. This reduces the computational requirement needed for pretraining allowing us to use larger sequence lengths.

## II. METHODS

Our proposed method can be broken up into two stages: a pretraining stage and a fine-tuning stage. During the pretraining stage, we train a model on our proposed objective without labels. During the fine-tuning stage, we select a down-stream task (i.e. in-hospital mortality) and train our model. In this section we describe our proposed pretraining objective. We start by describing our model architecture. We then describe our token and sequence level tasks. Finally, we describe data augmentation which is crucial for sequence level pretraining. A visual depiction of our pretraining can be seen in Figure 1.

We will define the notation used throughout this section. Let us denote a triplet as $(t, v, f)$ where $f \in \mathbb{N}$ is the index of some embedding associated with a measurement, $v \in \mathbb{R}$ is the value of the measurement, and $t \in \mathbb{R}_{\geq 0}$ is the time of the measurement. Let $\mathbf{x}_i = \{(t_j, v_j, f_j)\}_{j=1}^{T}$ be a sequence of triplets for an ICU stay indexed by $i$ where $T$ is the length of the sequence. Finally, let $\{\mathbf{x}_i, y_i\}_{i=1}^{N} \in \mathcal{D}$ be a dataset where $N$ denotes the size of the dataset, $\mathbf{x}_i$ is a sequence of triplets, and $y$ is a label associated with the sequence. Let $E$ represent a DNN called an encoder. This encoder takes $\mathbf{x}$ as an input and produces an embedding, $\mathbf{z}$.

### A. Model Architecture

In this section, we describe the model architecture used for our proposed pretraining. This architecture consists of 2 parts: an embedding layer and a backbone.

In this work we decide to use a triplet embedding first described in [9]. The triplet embedding works by learning a separate embedding for the value, measurement and time. Specifically, the value of the triplet is embedded using a single linear layer. For the measurement embedding, a lookup table to used to index the embedding. Unlike [9], we use a sinusoidal embedding [16] for our time value as we notice that it performs better. Additionally, following [3], [8], we use a learned class token which is appended to the end of a sequence.

We believe that there are many advantages to using a triplet embedding. (i) Unlike [13], we do not need to impute missing values. We believe that imputing missing values can inject human bias into the model. (ii) We do not need any special methods for dealing with the irregularly sampled data that is found in EHR time series. (iii) By using a triplet embedding, we are able to treat each measurement as a token similar to what is found in Natural Language Processing. This last point

is important for the formulation of our masked pretraining objective.

For the backbone of our model, we decide to use a transformer encoder [2] with causal self-attention. We propose using this model due to it's ability to handle long sequences of data.

### B. Sequence Level Task

Given a random data augmentation, $\mathcal{A}$, we can randomly augment a single input to create two positive pairs $\mathcal{A}(\mathbf{x}_i)$, $\mathcal{A}'(\mathbf{x}_i)$. Contrastive pretraining utilizes the InfoNCE objective to then learn to maximize the similarity between the embeddings of these two augmentations:

$$
\mathcal{L} = -\frac{1}{N} \sum_{i=1}^{N} \ln \frac{\exp(E(\mathcal{A}(\mathbf{x}_i))^T E(\mathcal{A}'(\mathbf{x}_i))/\tau)}{\sum_{j=1}^{N} \exp(E(\mathcal{A}(\mathbf{x}_i))^T E(\mathcal{A}'(\mathbf{x}_j))/\tau)}
$$
$$
-\frac{1}{N} \sum_{j=1}^{N} \ln \frac{\exp(E(\mathcal{A}'(\mathbf{x}_j))^T E(\mathcal{A}(\mathbf{x}_j))/\tau)}{\sum_{i=1}^{N} \exp(E(\mathcal{A}'(\mathbf{x}_j))^T E(\mathcal{A}(\mathbf{x}_i))/\tau)} \quad (1)
$$

Where $\tau$ is the temperature hyperparameter which controls the sharpness of the softmax distribution.

The difficulty lies in computing an unbiased estimator of this loss. In [8], [11], [12], the loss is computed for a batch of data, which may reduce computational cost. However, the summation in the denominator, referred to as the contrastive term in Eq 1, does not account for all negative pairs. Additionally, since the natural logarithm is a non-linear function, this results in a biased estimator of the gradient. To address this, large batch sizes, which require multiple GPUs, are used to obtain better estimations of the contrastive term. The challenge is further compounded in clinical data, where sequences of ICU measurements can be very long, necessitating additional computational resources.

Instead, we propose the use of a method from stochastic compositional optimization [17] where we develop a variance reducing estimator for the contrastive term. Given a batch of data $\mathcal{B}$, we let the contrastive term be define as $g(\mathbf{x}_i, \mathcal{A}, \mathcal{B}) = \sum_{j=1}^{|\mathcal{B}|} \exp(E(\mathcal{A}(\mathbf{x}_i))^T E(\mathcal{A}'(\mathbf{x}_j))/\tau)$. We define an estimator for the contrastive term as follows:

$$
\mathbf{u}_{i,t} = (1 - \gamma)\mathbf{u}_{i,t-1}
$$
$$
+ \gamma \frac{1}{2|\mathcal{B}|}(g(\mathbf{x}_i, \mathcal{A}, \mathcal{B}) + g(\mathbf{x}_i, \mathcal{A}', \mathcal{B})) \quad (2)
$$

Where $\gamma \in (0, 1)$ is a hyperparameter. We can then compute a stochastic gradient using our estimator instead of the contrastive term:

$$
\mathbf{m}_t = -\frac{1}{|\mathcal{B}|} \sum_{i=1}^{|\mathcal{B}|} \nabla(E(\mathcal{A}(\mathbf{x}_i))^T E(\mathcal{A}'(\mathbf{x}_i)/\tau)
$$
$$
+ \frac{1}{2|\mathcal{B}|\mathbf{u}_{i,t}}(\nabla g(\mathbf{x}_i, \mathcal{A}, \mathcal{B}) + \nabla g(\mathbf{x}_i, \mathcal{A}', \mathcal{B})) \quad (3)
$$

As the moving average updates, it incorporates more information about previous examples, making it a closer approximation of the true contrastive loss over the dataset. Hence, it

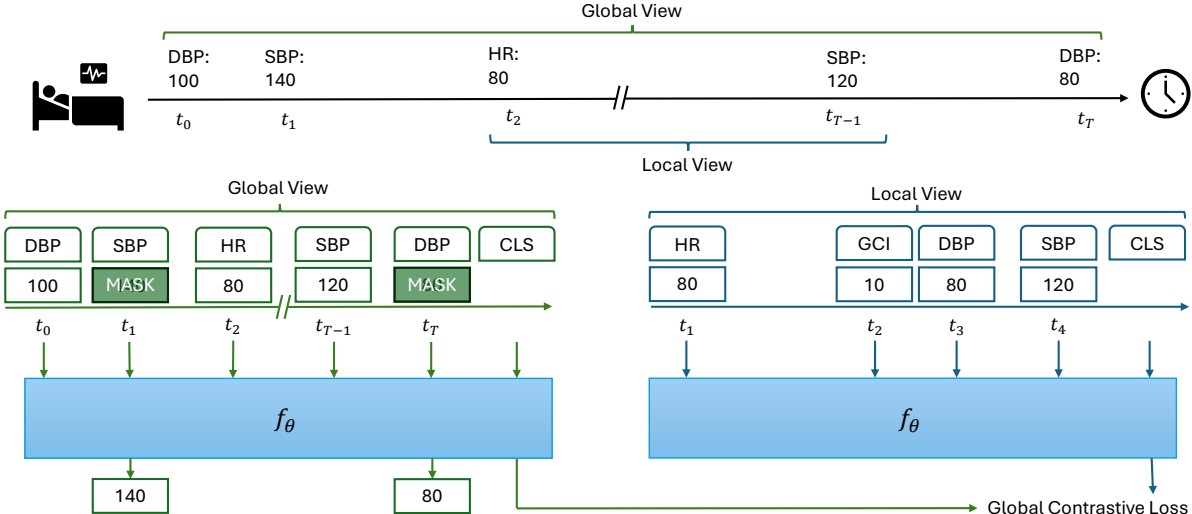

Fig. 1: We depict a single ICU stay using our propose pretraining method. A global (green) and local (blue) view of the time series are passed to our target model. Values from the global view are then masked with a random probability. The model trains to predict the masked values while learning to align the sequence level representations.

is termed the Global Contrastive Loss (GCL). A summary of this update is in Algorithm 1.

---

**Algorithm 1** Optimizing the Global Contrastive Loss

---

Set $\mathbf{u}^0 = 0$ and initialize $\mathbf{w}$
for $t = 1, \ldots, T$ do
    Sample a batch $\mathbf{B}$
    for $x_i \in \mathbf{B}$ do
        Compute $g(\mathbf{x}_i, \mathcal{A}, \mathcal{B})$ and $g(\mathbf{x}_i, \mathcal{A}', \mathcal{B})$
        Update $u_{i,t}$ according to Eq. 2
    end for
    Compute $\mathbf{m}$ according to Eq 3
    Update $\mathbf{v}_t = (1 - \beta_1)\mathbf{v}_{t-1} + \beta_1 \mathbf{m}$
    Update $\mathbf{w}_{t+1} = \mathbf{w}_t - \eta_1 \mathbf{v}_t$ (or Adam-style)
end for

---

*C. Triplet Level Task*

As described above, we propose treating each of the triplets as a token, similar to model pretraining in NLP. We would like to perform masked pretraining on these token but we believe that masking the entire triplet does not provide the model with enough information on how to reconstruct the masked segment. We instead decide to mask only the value of the triplet. In doing so, the model is aware of what type of measurement it is trying to reconstruct and at what time.

During pretraining we mask the triplet value using a trainable learned mask token. These triplets are embedded using the triplet embedding and passed to the backbone model. A linear layer uses the outputs of the backbone to make a prediction about the real value of the masked token. Since our masked token prediction is a regression task, we utilize mean-squared error to train our model. Specifically, given a set of masked indices $\mathcal{M}$ we use the following loss:

$$\mathcal{L}_{mask} = \frac{1}{|\mathcal{M}|} \sum_{i,j \in \mathcal{M}} \left(E(\mathcal{A}(\mathbf{x}))_{i,j} - v_{i,j}\right)^2 \qquad (4)$$

An advantage of training our model using this masked objective is that we are able to query a sequence for missing values by providing the model with a query time and measurement. We test the ability of our model to perform these queries in our experiments.

*D. Data Augmentation*

As described in [12], contrastive pretraining relies on a set of random augmentations tailored for the downstream task. These random augmentations create positive pairs, which can be thought of as two views of the same event in our time series data. We employ a multi-view augmentation approach as outlined in [12]. This approach involves selecting two distinct windows from an input sequence: a larger window termed the "global view," and either a perturbed version of this global view or a "local view." The goal of this augmentation strategy is to help the model effectively align detailed, fine-grained information with the broader, global aspects of the sequence. The selection between these views is performed randomly.

In the first method, random Gaussian noise is added to the values of the selected view. Employing this technique during training not only aids in capturing the global perspective but also enhances the model's resilience to noise.

The second method involves sampling the local view either by randomly choosing multiple small local regions or by selecting tokens from a single region. Our investigation revealed that both of these methods offer effective augmentations, enabling the model to discern and learn meaningful patterns.

### III. EXPERIMENTS

In this section, we test the quality of our pretraining. We start with settings used for understanding the learned

TABLE I: We report the results of our pretraining method along with several baseline methods with different percentages of labels. Results are the mean and standard deviation of 5 runs. % refers to the percentage of labels available during training.

| Model | % | In-Hospital Mortality | | Phenotype | | Imputation | |
|---|---|---|---|---|---|---|---|
| | | AUC-ROC | AUC-PR | Macro AUC-ROC | Micro AUC-ROC | MSE | MAD |
| LSTM | 100 | 0.839 (0.006) | 0.453 (0.015) | 0.755 (0.001) | 0.808 (0.001) | - | - |
| Baseline | 1 | 0.685 (0.031) | 0.253 (0.033) | 0.646 (0.008) | 0.737 (0.006) | - | - |
| | 5 | 0.789 (0.030) | 0.364 (0.031) | 0.714 (0.004) | 0.780 (0.003) | - | - |
| | 100 | 0.834 (0.017) | 0.462 (0.043) | 0.776 (0.003) | 0.823 (0.003) | 2.264 (0.629) | 1.180 (0.235) |
| STraTS [9] | 1 | 0.603 (0.104) | 0.185 (0.077) | 0.619 (0.013) | 0.724 (0.004) | - | - |
| | 5 | 0.771 (0.028) | 0.328 (0.027) | 0.704 (0.008) | 0.771 (0.006) | - | - |
| | 100 | 0.848 (0.004) | 0.455 (0.010) | 0.771 (0.004) | 0.820 (0.004) | 1.234 (0.016) | 0.630 (0.001) |
| GCL | 1 | 0.753 (0.021) | 0.299 (0.034) | 0.663 (0.012) | 0.748 (0.005) | - | - |
| | 5 | 0.817 (0.006) | 0.382 (0.020) | 0.717 (0.002) | 0.783 (0.002) | - | - |
| | 100 | 0.854 (0.002) | 0.459 (0.011) | 0.772 (0.004) | 0.820 (0.003) | - | - |
| Masked | 1 | 0.724 (0.030) | 0.301 (0.025) | 0.634 (0.013) | 0.732 (0.005) | - | - |
| | 5 | 0.800 (0.021) | 0.370 (0.029) | 0.707 (0.010) | 0.775 (0.006) | - | - |
| | 100 | 0.854 (0.004) | 0.468 (0.007) | 0.776 (0.001) | 0.823 (0.001) | 0.439 (0.025) | 0.360 (0.004) |
| Combined | 1 | 0.751 (0.016) | 0.310 (0.027) | 0.660 (0.013) | 0.745 (0.007) | - | - |
| | 5 | 0.811 (0.010) | 0.394 (0.011) | 0.723 (0.002) | 0.783 (0.002) | - | - |
| | 100 | 0.852 (0.017) | 0.462 (0.043) | 0.773 (0.030) | 0.821 (0.003) | 0.409 (0.024) | 0.351 (0.001) |

representations from contrastive pretraining: linear and semi-supervised evaluation. We then evaluate the effects of masked pretraining by measuring the error between imputed and ground truth values. Next, we evaluate the performance of our pretraining when the number of measurement features is scaled. Finally, we evaluate the ability of our model to transfer.

We use a 2 layer transformer network with a triplet embedding. We use the PyTorch [18] library to create our experiments. We use a single NVIDIA GeForce GTX 1080 Ti for evaluation experiments while 8 are used for pretraining to allow for large batch sizes. Our code is available on GitHub at https://github.com/shiveshchowdary/EHR-ContrastiveLearning. The model can be downloaded on HuggingFace at https://huggingface.co/Shivesh2001/EHR-CombinedModel-MIMIC

### A. Data

We utilize two open-source ICU datasets: the MIMIC-III database [19] and the eICU dataset [20]. The MIMIC-III database includes various measurements from thousands of ICU stays. The eICU dataset contains similar ICU time series data but spans a more diverse range of hospitals across the United States. We simulate pretraining a model on a large academic hospital's data (MIMIC-III) and then transferring it to more diverse national data sources (eICU).

We follow the exclusion criteria from Harutyunyan et al. [13] to remove patients with ICU transfers, pediatric patients, and those with multiple ICU stays per hospital admission. For pretraining, we use 69 measurements, but for benchmark tasks, we only use the 17 features outlined in the MIMIC-III benchmark. The 69 features used during pretraining include the 17 features used during downstream tasks. Similar to Harutyunyan et al. [13], we remove outliers, one-hot encode categorical features, and standardize the data before training. For the eICU data, we standardize the data using the mean and standard deviation of the MIMIC-III

features. A full list of the features used can be found in the code for this paper. This demonstrates our model's flexibility to handle any combination of measurements without needing retraining when only a subset of features is available.

We divide each dataset into training, validation, and test splits consisting of 70%, 15%, and 15% of the total data respectively. Hyperparameter tuning is conducted by training on the training split and evaluating on the validation split, with final results reported from the test split.

### B. Pretraining

We pretrain our model using our proposed pretraining objective for 400 epochs using an adam optimizer with de-coupled weight decay [21]. We use a cosine learning rate schedule with a linear warmup of 5 epochs. We use a weight decay of $1e^{-5}$ and a batch size of 512. We tune the temperature in $\{0.01, 0.03, 0.07, 0.01\}$, the learning rate in $\{1e-4, 1e-3, 1e-2\}$, and $\gamma$ in $\{0.1, 0.3, 0.5, 0.7, 0.9\}$. We use the linear evaluation objective to evaluate our pretrained models performance. In our experiments, we distribute the pretraining over 8 GPUs

### C. Downstream Tasks

We would like to understand how well a model pretrained using our objective will do on some useful downstream tasks. We decide to use two common benchmark tasks [13] for comparison with other methods. Those tasks include in-hospital mortality, and phenotyping. We provide a description of those tasks along with the metrics used with each one below:

1) **In-hospital mortality**: Given the first 48 hours of measurements, we measure how well our model can predict whether a patient will live or expire at the end of their ICU stay. This is a binary classification task so we use binary cross entropy to train. For this task, we use AUC-ROC and AUC-PR to measure performance.

2) **Phenotyping**: Given a time series of measurements, predict which of the 25 phenotypes are present. This is a multi-label classification task so we use binary cross entropy for this task. For this task, we use micro-AUC-ROC and macro-AUC-PR to measure performance.

These two task evaluate the ability of a model to learn sequence level information or tasks that require information from the entire sequence. We also include an additional task which evaluates the ability of the model to complete token level tasks. We evaluate the ability of our pretrained model to impute missing values. In this task, we randomly mask measurement values and measure the reconstruction error between the models prediction and the true value. In doing so, we evaluate how well a model can impute a missing value given a query time and measurement from an existing sequence. We measure the reconstruction error as the Mean Squared Error (MSE) and the Mean Absolute Error (MAE)

### D. Comparisons

In all of our experiments, we compare our method to several baseline methods. We include an LSTM model used by [13] which is trained on data containing all possible measurements at each time step. Since not all measurements are taken at each time step, this method imputes missing values using recent measurements or using normal measurement values. Additionally, the time series data is averaged over each hour producing uniform time steps. We include this comparison to test the efficacy of our triplet embedding.

We include a related pretraining method which proposes the use of forecasting as a pretraining task [9]. In this method, a portion of the measurement values at the end of the time series are masked. The model is then trained to reconstruct these values using the rest of the sequence. We replicate this pretraining with our model and features. Since we mask 10% of our input data at random, we apply a mask to the last 10% of the measurements for the forecasting task.

We include 3 baselines for comparison with our method. The first, which we call the baseline method, is a randomly initialized transformer model with the same architecture as the pretrained models. This method is included to understand the effects of pretraining. We also include a contrastive, which we call GCL, and masked method which utilize Equation 3 and Equation 4 respectively. We train each of these methods using the same training recipe as our combined method. These methods act as an ablation study of our pretraining method.

### E. Semi-Supervised Evaluation

Semi-supervised evaluation have been used to evaluate the quality of the representations learn by JE-SSL methods. This evaluation utilizes all the available input data for a pretraining. Afterwards, a randomly initialized linear classifier is trained along with the pretrained model using only a fraction of the available labels on a downstream task. Intuitively, if the pretraining process has learned useful features for our target downstream task, then fewer labels will be necessary to achieve desired results. More importantly, we believe that, if

successful, our method will help reduce the cost of producing labels, freeing up both time and resources for clinics.

For our semi-supervised experiments, we use an Adam optimizer [22] a cosine learning rate scheduler [23]. Following [14], we use a different learning rate for the linear layer and the backbone layer for pretrained models to avoid forgetting information learned during pretraining. We conduct a grid search on the linear layer learning rate between $\{0.001, 0.01, 0.1\}$, the backbone learning rate in $\{1e-4, 5e-4, 8e-4, 1e-3\}$, the number of epochs $\{2, 4, 6, 10, 20, 50\}$, and the batch size in $\{8, 16, 32\}$. We use early stopping with a patients of 5 epochs without improvement in the validation loss.

We report the results of our experiments wit 1% and 5% labeled data in Table I along with 100% of the labeled data for comparison. We first note that almost all methods perform similarly when 100% of the data is available with the exception of the LSTM model which utilizes imputed data. We see that even our baseline model outperforms this method, indicating that our proposed architecture and triplet embedding perform better than data imputation methods.

When comparing semi-supervised experiments, we see that each of the pretraining methods performs better than the randomly initialized baseline. As the number of labels decreases, we see an increase in the performance different between the pretrained methods and the baseline. We also see that both of the contrastive methods perform best indicating that sequence level pretraining performs best on these tasks.

### F. Linear Evaluation

The linear evaluation task tests the ability of JE-SSL methods to learn meaningful representations for downstream tasks. In this method, the parameters of the backbone model are not trained. A linear layer is then randomly initialized. Using the final layers class token output, the linear model is then trained on the downstream task. Intuitively, if the model has learned useful feature during pretraining, then simple linear decision boundaries can be drawn between classes.

We perform this experiment for our 2 sequence level downstream tasks using our proposed loss. For comparison, we report the results of a randomly initialize model as the baseline. We report the the results of contrastive pretraining, or SimCLR, without the moving average estimator used in our proposed method as a comparison. We also perform an ablation study on our proposed pretrained method by pretraining a model using the masked and contrastive objectives separately. We train linear classifier using an Adam optimizer [22] with a learning rate of 0.1 and a cosine learning rate scheduler [23]. We report the results in Table II.

Surprisingly, our baseline method achieves a non-random prediction, as indicated by an AUC-ROC greater than 0.5. Among the pretrained models, the contrastive methods perform the best with GCL methods performing better than SimCLR. This is expected, as contrastive pretraining is a sequence-level task, which aligns better with our sequence-level evaluation compared to token-level tasks like masked pretraining. Additionally, the ability of GCL to access information from

TABLE II: We report the results of a linear evaluation experiments on the in-hospital mortality and phenotyping benchmarks as the mean and standard deviation of 5 random runs. The "Pretrain Features" column indicates the number of features that were used during the pretraining phase. During the finetuning phase, each task was evaluated using 17 features.

| | | In-Hospital Mortality | |
| Model | Pretrain Features | AUC-ROC | AUC-PR |
|---|---|---|---|
| Baseline | 69 | $0.642 \pm 0.001$ | $0.195 \pm 0.001$ |
| SimCLR | 69 | $0.802 \pm 0.004$ | $0.348 \pm 0.013$ |
| GCL | 69 | $0.817 \pm 0.001$ | $0.399 \pm 0.001$ |
| Masked | 69 | $0.722 \pm 0.001$ | $0.264 \pm 0.001$ |
| Combined | 17 | $0.713 \pm 0.010$ | $0.280 \pm 0.012$ |
| Combined | 69 | $0.814 \pm 0.001$ | $0.399 \pm 0.001$ |

| | | Phenotype | |
| Model | Pretrain Features | Macro AUC-ROC | Micro AUC-PR |
|---|---|---|---|
| Baseline | 69 | $0.607 \pm 0.001$ | $0.718 \pm 0.001$ |
| SimCLR | 69 | $0.706 \pm 0.004$ | $0.764 \pm 0.025$ |
| GCL | 69 | $0.718 \pm 0.001$ | $0.783 \pm 0.001$ |
| Masked | 69 | $0.644 \pm 0.001$ | $0.735 \pm 0.001$ |
| Combined | 17 | $0.654 \pm 0.010$ | $0.734 \pm 0.026$ |
| Combined | 69 | $0.712 \pm 0.001$ | $0.777 \pm 0.001$ |

other batches allows it to optimize an objective that is closer to the true unbiased contrastive objective. This shows that our proposed contrastive objective and augmentation strategy are capable of learning useful features for these downstream tasks.

### G. Measurement Imputation

In the previous section, we demonstrated that combining masked pretraining with contrastive learning yields superior results. We now aim to evaluate the impact of our masked pretraining on triplet-level tasks. To do this, we simulate a "what-if" scenario where a clinician has a time series of patient measurements and wants to query another specific measurement at a given time. For each batch, we randomly mask measurement values using our masked token and pass the sequence through our pretrained model. Training for this scenario occurs during the pretraining phase, so no additional updates are required for this evaluation. We compare our proposed method to both masking alone and forecasting. The results of these experiments are presented in Table I.

Our method achieves results comparable to masking. However, while the masking method under-performs on sequence-level tasks, our method maintains similar performance while also effectively learning sequence-level information.

*1) Data Scaling Properties:* One of the advantages of our proposed framework is ability of our model to handle any number of features without the need for retraining. During pretraining, we utilized 69 features. We would like to understand how the number of features used during pretraining will affect the ability of our model to make predictions on down stream task. We pretrain a model using only the 17 features utilized in our down stream tasks. We note that by reducing the number of features in this way, we found that 116 ICU stays did not contain any measurements, reducing the number of training examples. We compare our model pretrained on 69 features with our model pretrained on 17 features using a linear evaluation on our two sequence level tasks. Each of these evaluations only uses the 17 features outlined in the MIMIC-III benchmark paper. The results are in Table II.

While each of the pretrained models performs better than the baseline, we see that the model pretrained with fewer measurements performs significantly worse that the model pretrained with more features even when the same number of features are used for the down stream task. This indicates that our model, pretrained with our proposed objective, are learning valuable information from additional features that aren't present with a smaller feature subset.

### H. Transferability

We have seen that our pretraining objective results in good representations for downstream tasks as indicated by our linear evaluation experiments. We now ask if these representations are robust enough to transfer to other hospitals. We evaluate our model on the eICU dataset using linear evaluation.

TABLE III: We transfer our pretrained model to an external source of data, eICU dataset, and evaluate the performance using the linear evaluation protocol. We report the results as the mean and standard deviation of 5 random initializations.

| | In-Hospital Mortality | |
| Model | AUC-ROC | AUC-PR |
|---|---|---|
| Baseline | 0.633 (0.618, 0.647) | 0.156 (0.143, 0.170) |
| GCL Pretraining | 0.775 (0.774, 0.776) | 0.255 (0.254, 0.256) |
| Masked Pretraining | 0.655 (0.654, 0.656) | 0.135 (0.134, 0.136) |
| Combined Pretraining | 0.744 (0.743, 0.745) | 0.231 (0.230, 0.232) |

| | Phenotype | |
| Model | Macro AUC-ROC | Micro AUC-PR |
|---|---|---|
| Baseline | 0.591 (0.584, 0.597) | 0.734 (0.731, 0.737) |
| GCL Pretraining | 0.690 (0.689, 0.691) | 0.790 (0.789, 0.791) |
| Masked Pretraining | 0.613 (0.612, 0.614) | 0.750 (0.749, 0.751) |
| Combined Pretraining | 0.683 (0.682, 0.684) | 0.784 (0.783, 0.785) |

The results show that the two contrastive methods perform best when evaluated on the eICU dataset. This indicates that these methods are particularly effective at learning robust features that generalize well across different datasets. Specifically, the success on the eICU dataset, which encompasses a more diverse range of hospitals and patient populations across the United States, highlights the ability of these contrastive methods to transfer learned features to varied and heterogeneous settings. This robustness is crucial for developing models that can be applied broadly in clinical practice, ensuring reliable performance across different hospital environments and patient demographics. By learning from diverse and representative

data during the pretraining phase, the model can better handle variations and complexities in new, unseen data.

## IV. LIMITATIONS

During pretraining, we utilize the linear evaluation on the in-hospital mortality benchmark to select the best pretrained model. However, this method of evaluation has several limitations. The first being that this method requires the training of a randomly initialized linear layer. The initialization affect the outcome of the evaluation. Additionally, it is not clear how to select the hyperparameters for training this linear classifier.

Lastly, this evaluation relies on a good downstream task. In the case of ICU time series data, there could be numerous tasks that could be selected including sequence level and token level tasks. It is unclear which task should be used for this evaluation. Future works could investigate better methods for evaluating these model during pretraining.

## V. DISCUSSION AND CONCLUSION

In this paper, we have proposed a novel method for pretraining long EHR time series data which combines masked imputation with contrastive learning. We evaluated the ability of a model pretrained using our objective to learn meaningful representations for clinical downstream tasks using a linear and semi-supervised evaluation. The results of our semi-supervised evaluation showed that our proposed method is capable of enhancing training where few labels are available. We believe that this is an important step for clinics that would like to take advantage of their EHR databases but do not have the time or resources to produce significant amounts of labeled data.

We further tested the ability of our model to impute missing data. We simulated a "what-if" scenario where our model was queried for measurement values at a predefined measurement and time. We saw that our pretrained model was capable of achieving results close to their ground truth values making our pretraining a useful tool for gaining insight about a patient without the need for invasive measurements.

We test the ability of models pretrained using our objective to scale in performance as the number of measurement features increased in our pretraining dataset. We saw that our model is capable of handling any number of measurements and actually increases in performance when more measurements become available. This coupled with the flexibility of the triplet embedding allows our model to be transferred to task that use any subset of the pretraining features.

After evaluating the quality of learned representations, evaluate the transferability of our pretrained model to other domains by evaluating it on eICU. We saw that while our model does not perform as well on this dataset, it still learns meaningful representations which we believe is a step forward towards transferable foundation models for EHRs.

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
