# OpenReview forum: "An Efficient Contrastive Unimodal Pretraining Method for EHR Time Series Data"
_IEEE.org/EMBS/BHI/2024/Conference — IEEE BHI'24_

### Official Review · Reviewer_LH66 · 2024-07-21
**Revision round 1**

**Overall Rating:** 5
**Confidence:** 4

**Review:**

Overall, the manuscript is very extensive in terms of text descriptions, which can contribute to a clearer exposure of all the details and considerations followed by the authors to explain their complete method, but also increase the risk for more text typos, unclear text passages and reduction of the presented performance results in favor of the accompanying explanatory text. Moreover, the impact and significance of the current study are also diminished by the many limitations found by the authors during the validation and testing of their method. Additional comments are provided below.

**Other Quality Metrics:**

(a) Clarity of writing: fair
(b) Clinical significance: fair
(c) Methodological novelty: good
(d) Experiments and results: fair

**Questions For The Authors:**

1 - Section I. Introduction: the following sentence is not understandable: "... We see a steady increase in model performance when both setting are scaled measured using the linear evaluation..."
2 - Section III. Methods: missing text when referring to the DNN encoder with "x" input and "z" outputs. Please correct.
3 - Section IV. Experiments: authors mention the computational platform for experiment evaluation, but not the involved computational workload or execution times.
4 - Section IV. Experiments. A. Data: What is the percentages for the split training, validation and test stages?
5 - Section IV. Experiments. C. Downstream task: the following sentence is not understandable: "... in this task, we randomly mask measurement values measure the reconstruction error between the models and the true value ..."
6 - Section IV. Experiments. H. Transferability: authors claim that the two contrastive methods perform better in the eICU dataset. But from the values in Table III, they are lower than those presented in Table II. Moreover, the last paragraph in the "Discussion and Conclusion" section seems to contradict exactly the initial statement of the authors regarding this topic.

**Strengths:**

1 - The presence of a "Related Works" section
2 - Description of authors' choices (and advantages) relative to other published studies in literature for EHR modeling and joint embedding self-supervised learning.

**Summary Of The Paper:**

The present manuscript proposes a novel method for pretraining long EHR time series data, combining masked imputation methods with contrastive learning, in an attempt to learn meaningful representations for clinical downstream tasks.

**Weaknesses:**

1 - Small number of literature references to compare with authors' work.
2 - Considerable number of pre-processing tasks still performed on the datasets before training.

---

> ### Author Rebuttal · Authors · 2024-09-02
>
> *Weaknesses*
> 1. Unfortunately, we agree! The field of EHR measurement time-series pretraining hasn’t received much attention. We include forecasting from the STraTS in our table as a comparison to our work. In the final version, we will add the citation here to further clarify the scope of related work, our work, and what we know to be the state of the art in the field at the time of this writing.
>
> *Questions*
>
> 1. We demonstrate in our linear evaluation results in Table II that we are able to scale the number of types of measurements in our dataset and increase the performance of our pretrained model. We have changed this sentence to the following: “We see an increase in model performance as the number of measurement features increases as measured by our linear evaluation.”
>
> 2. “We changed this section to say the following: “Let E represent a DNN called an encoder. This encoder takes x as an input and produces an embedding, z.”
>
> 3. We performed an additional experiment pretraining our model and that pretraining takes 4 hours and 26 minutes with a memory size of 3318 MB across each GPU.
>
> 4. The train, validation, and test split percentages are 70%, 15%, and 15%. We will clarify this information in our data section.
>
> 5. Thank you for pointing out this mistake. There is a missing “and” in this sentence. The sentence should read as follows: "... in this task, we randomly mask measurement values and measure the reconstruction error between the models and the true value ..." Here we are removing or masking measurement values and determining how well our model can reconstruct those missing values.
>
> 6. We apologize for the confusion. Here when we state that the two contrastive methods perform better on the eICU dataset, we are comparing GCL and Combined pretraining to Masked and the baseline on eICU. We do not compare the results of our linear evaluation on eICU to our results on MIMIC-III. We will change the first sentence of the second paragraph of the transferability section to the following for a clearer explanation: “The results show that the two contrastive outperform the masked pretraining and baseline method in the eICU setting.”

---

### Official Review · Reviewer_FrqP · 2024-07-25
**An Efficient Contrastive Unimodal Pretraining Method for EHR Time Series Data**

**Overall Rating:** 6
**Confidence:** 5

**Other Quality Metrics:**

(a) Clarity of writing: excellent
(b) Clinical Significance: good
(c) Methodological Novelty: good
(d) Experiments and Results: good

**Questions For The Authors:**

1. Given that a primary contribution of this work is stochastic compositional optimization, could you conduct additional experiments to examine the impact of batch size and batch selection?
2. On Page 3, Formula 1, please provide more detailed explanations of the terms $\mathcal{A}$ and $E$.
3. The code should be made open-source upon acceptance.
4. Typographical error. Please check and correct the use of ' " ' in the main text.
5. In Table 1, clarify the meaning of "%" in the caption.
6. Describe how you tuned the hyper-parameters in your experiments (e.g., grid search, random search, or other methods) in the main text.

**Strengths:**

1. Fascinating topic. Addressing the issue of missing values in EHR datasets is currently a highly discussed subject.
2. Comprehensive experiments. The research includes experiments on two different datasets under various conditions.
3. Excellent writing. The paper is exceptionally well-written and easy to understand.

**Summary Of The Paper:**

The paper presents an innovative approach for pretraining deep neural networks using extensive clinical time series data from electronic health records (EHRs). This method combines contrastive learning with masked token prediction to effectively extract significant features from the data.

**Weaknesses:**

1. Marginal improvement. The proposed method appears to offer only a slight improvement in the results.
2. Lack of statistical testing. The author did not demonstrate that the proposed method is statistically significantly better than others.

---

> ### Author Rebuttal · Authors · 2024-09-02
>
> 1. We are unable to run batch sizes large than the one used in our experiments due to computational constraints. However, we report the linear evaluation results of SimCLR to our method in table 6. This method is very similar to ours. However, it does not contain the stochastic compositional optimization proposed in our paper. We saw that in our experiments, our method provided an increase in performance on the Phenotyping task.
>
> 2. In this section, A represents a random augmentation of the data. An example of this for images is a random crop, rotation, or adding gaussian noise. We describe the augmentations that we use for our measurement time-series data in Section III Subsection D.
>
> 3. We completely agree and already plan for code for our experiments tol be released to GitHub. Additionally, model weights for our pretrained model will be released to HuggingFace.
>
> 4. We will remove this in the final version of the paper, thank you.
>
> 5. The “%” symbol in Table 1 refers to the percentage of labels used for finetuning. We agree that this isn’t clear to readers. We will add the following description to the caption of Table 1: “‘%’ refers to the number of labels used to fine tune the model.”
>
> 6. We conduct a grid search on the hyperparameters outlined in the Experiments section. We apologize for not stating this beforehand. We will add this detail to our final print ready version.

---

### Official Review · Reviewer_m6vz · 2024-08-09
**An Efficient Contrastive Unimodal Pretraining Method for EHR Time Series Data**

**Overall Rating:** 6
**Confidence:** 4

**Other Quality Metrics:**

(a)	Clarity of writing: Good (The paper could benefit from more detailed explanations of the proposed method and experimental design.)
(b)	Clinical significance: Great (The paper addresses a significant problem in healthcare, and the proposed method has the potential to improve the analysis of EHR time series data.)
(c)	Methodological novelty: Great (The combination of sequence-level and token-level tasks is novel, but the paper could benefit from more detailed explanations of the approach.)
(d)	Experiments and results: Good (The evaluation of the method is limited, and the results could be more thoroughly discussed.)

**Questions For The Authors:**

1. How does the performance compare to other recent self-supervised methods for time series data?
2. How sensitive is the method to hyperparameters like the contrastive temperature?
3. Have the author explored using this pretraining approach with other model architectures beyond transformers?
4. In the experiments section, please give a more detailed description of the data, especially the two down-stream tasks: i.e., the incidence of mortality, the clinical meaning of the 25 phenotypes, and whether different phenotypes will co-occur on one patient.
5. Have the author thoroughly explored the pretraining performance when the percentage of data scaled? While the paper reports performance merits for 1%, 5%, and 100% data usage, a graphical depiction of performance with incremental data (e.g., each 5% or 10% increase) would indeed be helpful.

**Strengths:**

1.	Addresses important challenges in learning from EHR data - irregular sampling, missing values, limited labels
2.	Novel combination of token-level and sequence-level pretraining objectives
3.	Efficient approach to contrastive learning that can handle long sequences
4.	Comprehensive evaluation on multiple downstream tasks and datasets
5.	Demonstrates scalability and transfer learning capabilities

**Summary Of The Paper:**

This paper proposes a novel method for pretraining deep neural networks on electronic health record (EHR) time series data using self-supervised learning. The method aims to address the challenges of large labeled data requirements by utilizing self-supervised learning techniques. The authors introduce a modified triplet embedding approach that combines sequence-level pretraining tasks and token-level tasks, demonstrating the effectiveness of their method through various downstream tasks.

**Weaknesses:**

1. Limited comparison to other state-of-the-art self-supervised methods for time series
2. Limited exploration of different model architectures beyond transformers: the paper focuses solely on transformer architectures. Exploring the method’s performance with other architectures (e.g., RNNs, CNNs) would provide insights into its generalizability.
3. No discussion of computational requirements or training times: there is no discussion of computational requirements, training times, or memory usage. This information is crucial for assessing the method’s practicality in real-world settings, especially given the focus on efficiency.

---

> ### Author Rebuttal · Authors · 2024-09-02
>
> 1. We compare our method to Strats, a measurement time-series pretraining method that utilizes forecasting of future measurements. We reported the results of this model in the row labeled forecasting. However, we feel that this is not clearly labeled in our manuscript. We will replace this with the name of the method, STrATS [9], and the citation in the print ready version.
>
> 2. In our experiments, we did not see a significant difference in temperatures $\pm$ 0.01.
>
> 3. We pretrained an LSTM model and reported our semi-supervised results below. Performance on the IHM task with the LSTM model did not change significantly. However, using a transformer instead of the LSTM led to a significant improvement in the Phenotyping task. This may be because IHM task information is located at the end of the time series, while Phenotyping requires information from the entire sequence. LSTMs may forget early information in long sequences, making them less suited for this task.
>
> IHM RESULTS
> Task | Data Usage | LSTM AUC ROC | Transformer AUC ROC | LSTM AUC PR | Transformer AUC PR
> combined | 0.01 | 0.6991(0.0676) | 0.751(0.016) | 0.2399(0.0618) | 0.310(0.027)
> combined | 0.05 | 0.8100(0.0074) | 0.811(0.010) | 0.4066(0.0199) | 0.394(0.011)
>
> PHENOTYPE RESULTS
> Task | Data Usage | LSTM Macro | Transformer Macro | LSTM Micro | Transformer Micro
> combined | 0.01 | 0.5951(0.0159) | 0.660(0.013) | 0.7170(0.0051) | 0.745(0.007)
> combined | 0.05 | 0.6398(0.0144) | 0.723(0.002) | 0.7391(0.0088) | 0.783(0.002)
>
> 4. For the two down-stream tasks, we use two tasks from the MIMIC-III Benchmark paper. The Phentoyping tasking is a multi-label classification problem of 25 Phenotypes outlined in the MIMIC-II Benchmark. Patients can have none or multiple phenotypes for this problem. These phenotypes are different acute or chronic conditions that the patients are diagnosed with.
> Mortality in MIMIC-III occurs at a rate of 13.5% , 13.5%, and 11.5%  in the train, validation, and test spli. For the eICU dataset, mortality occurs at a rate of 9.0% for all splits.
>
> 5. While we agree that graphical representations show broader ranges, the number of experiments conducted, with the number of use cases and number of tasks, we felt a table better represented the results, unfortunately particularly with length limitations in the paper. We will add such a graphical representation to any supplementary material we are allowed to upload in the print ready versions.

---

### Decision · Program_Chairs · 2024-09-22

Accept